# Simulation-Based Considerations on the Rayleigh Criterion in Super-Resolution Techniques Based on Speckle Interferometry

Yasuhiko Arai * and Tong Chen

Department of Mechanical Engineering, Faculty of Engineering Science, Kansai University, 3-3-35, Yamate-cho, Suita 564-8680, Osaka, Japan
* Correspondence: arai@kansai-u.ac.jp

**Abstract:** This study sought to explain the physical phenomenon that eludes the constraints of the Rayleigh criterion in the microstructure observation method using speckle interferometry, for which super-resolution has been experimentally proven; the study was conducted using computer simulations. Separating the light from two light sources in close proximity, which exceeded the Rayleigh criterion under incoherent light, was believed to be impossible. The simulation results, however, showed that when coherent light is used, the separation of two close points is not necessarily impossible if the light phases between the two points are different. Furthermore, the resolution of microstructure observation techniques based on speckle interferometry was discussed. A new interpretation of the Rayleigh criterion in super-resolution techniques based on speckle interferometry was reported.

**Keywords:** Rayleigh criterion; super-resolution; coherent light source; speckle interferometry; computer simulation



## 1. Introduction

Super-resolution technology is an important observational technique that supports advances in biotechnology. When super-resolution technology is viewed from a broad viewpoint, it can be broadly divided into two categories: optical subjects that must be considered when collecting images of micro-objects that exceed the diffraction limit of the lens and subjects that must be considered when processing the collected images. The previous category is based on the Rayleigh criterion in observation optics, Abbe's theory of image formation, etc. [1,2]. The second category can be considered image processing techniques, such as the Lucy–Richardson method [3,4], which processes the collected images.

In these categories, this study discusses the problems of super-resolution, which has already been reported as a technique for observing microstructures beyond the diffraction limit, based on speckle interferometry.

Traditionally, microstructural observations have been performed using optical microscopy. However, it is widely acknowledged that optical microscopy cannot observe microstructures that exceed the diffraction limit of observation optics, as indicated by the Rayleigh criterion [1,2].

Several techniques have been proposed to avoid the Rayleigh criterion in order to achieve super-resolution. For example, in biotechnology, fluorescent proteins have recently been used to observe microstructures [5–15]. Specifically, new techniques, such as photoactivated localisation microscopy (PALM) and stimulated emission depletion (STED), have been developed to facilitate new biotechnological research. In addition, imaging of nanoscale objects has been achieved by bringing dielectric microspheres into contact with the subject [16,17]. Furthermore, new nanoscale observation techniques have been reported, such as superlens imaging [18] of objects several nanometres in size has been attempted.

Although image acquisition beyond the Rayleigh criterion is considered impossible in optical microscopy, if it were possible, it would be conceivable to capture moving images for

extended periods as well as high-speed photographs of dynamically active living organisms in two dimensions. If such techniques can be developed, image-capturing techniques using optical microscopy will become an attractive technology to support developments in bio-research. Therefore, the development of such technology is eagerly awaited.

Recently, a technique for observing the shape of microstructures beyond the diffraction limit, which analyses the phase of light based on speckle interferometry [19–21], has been reported [22–26]. This new observation technique achieves super-resolution by detecting the phase distribution of light from the observed object, instead of processing only information from images captured as a light intensity distribution, as in conventional techniques. Super-resolution is obtained by analysing the information from the viewed object as a phase distribution of light using the speckle interferometry method.

In this method, based on Abbe's image theory [27], scattered light with many ray vectors is used as illumination light to increase the number of rays passing through the lens aperture. The phase change at the confocal on the imaging element on the surface under test is reconstructed in the computer as a two-dimensional phase distribution, and the surface shape of the object under test is observed as a three-dimensional shape distribution.

However, this technique based on speckle interferometry has been able to achieve, albeit experimentally, observations that exceed the Rayleigh criterion, which for many years was thought to be unreachable. However, there is no clear explanation as to what kind of physical phenomena enable this observation technique to exceed the Rayleigh criterion.

In this study, physical phenomena that elude the constraints of the Rayleigh criterion in microstructure observation methods using speckle interferometry were explained with computer simulations using COMSOL Multiphysics [25], which is capable of electromagnetic field simulation analysis.

This study clarifies that the Rayleigh criterion [2], which is based on the analysis of the intensity distribution of light assuming traditional incoherent light, must take into account the phase variation between nearby light sources when dealing with coherent light.

## 2. Materials and Methods

### 2.1. Techniques for Observing Microstructures beyond the Diffraction Limit Based on Speckle Interferometry

In the light of the microstructure observation technique used in this study, for example, it is assumed that the cross section of the measured object shown in Figure 1a can be defined as f(x).

Based on this assumption, when a lateral shift δx is given to the measurement object, as shown in Figure 1b, the shape displacement occurring at each measurement point can be defined as f(x) − f(x + δx) from the speckle interferometric measurement method presented in a previous report [22]. The displacement of the shape is then accurately measured using speckle interferometry, and the pseudo-differential value {f(x) − f(x + δx)}/δx in the shift direction with respect to the shape is obtained by dividing the detected displacement by the lateral shift value. Furthermore, the shape of the measurement object can be reconstructed by integrating pseudo-differential values.

In the calculation process, the phase distribution obtained by integration is aligned in two dimensions based on the relationship between the positions of each confocal point (P'c) at each measurement point (Pc), as shown in Figure 1c, resulting in the reconstruction of a three-dimensional shape f(x).

If the Rayleigh criterion is exceeded at nearby measurement points, the microstructure cannot be observed according to the traditional idea. However, super-resolution has been experimentally performed using this method based on speckle interferometry under beyond the diffraction limit. Simulations were performed in this study to answer this question.

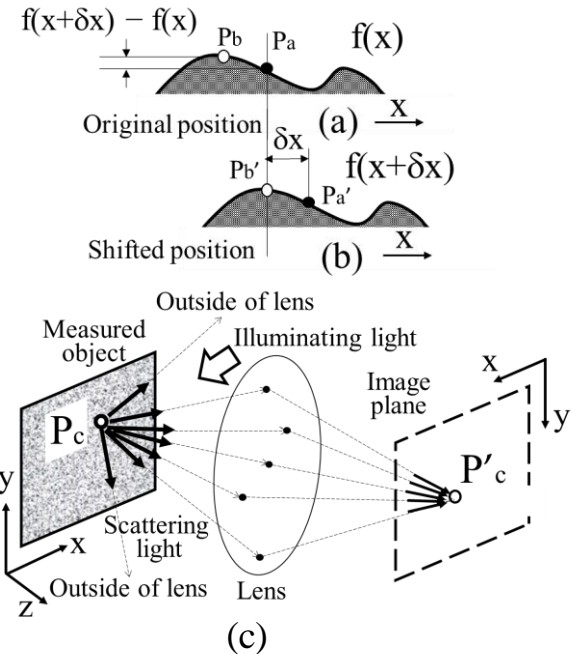

**Figure 1.** Principle of the measurement system. (**a**) Section of the measured object at the original position. (**b**) Section of the measured object at the shifted position. (**c**) Detection of two-dimensional phase distribution using the perfect optical system.

### 2.2. Simulation Model

As shown in a previous report [26], a speckle interferometer was constructed using a laser source with a wavelength of 532 nm, an objective lens (Mitutoyo M Plan Apo 200×) with a magnification of 200×, and an aperture (NA) of 0.62.

In speckle interferometry, only two speckle patterns are captured before and after the lateral shift of the measured object. Super-resolution images can be produced at a resolution of several tens of nanometers [23,24] using the speckle patterns.

However, it is extremely difficult to remove disturbances completely, such as the effects of stray light on the actual optical system, to confirm the principle of this method, as attempted in this study. In addition, it is difficult to discuss physical phenomena in detail owing to the limitations of measurement accuracy and the experimental environment.

This study investigates how electromagnetic simulation software (COMSOL Multiphysics) [25] can be used to observe microstructures beyond the diffraction limit.

The computer simulation model used in the study is shown in Figure 2a.

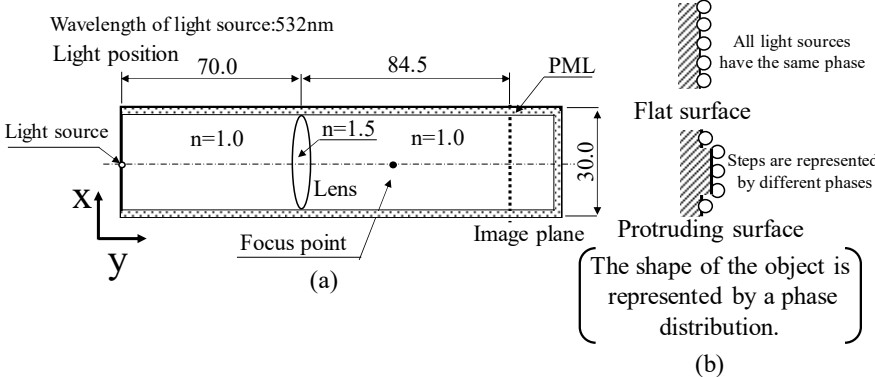

**Figure 2.** Computer simulation model. (**a**) Optical system. (**b**) Measured object.

In experimental optics, a reflective diffraction grating made of glass and a microstructure drawn on a silicon wafer using an EB lithography machine were used as the measured

objects. The light reflected from the measured object [22–26] was analysed. The simulation described in a previous paper [25] set up a sinusoidal shape on the glass surface as the measurement object model. However, because this study is concerned with the Rayleigh criterion, the measurement objects were further simplified.

In other words, when a flat plane is the measurement object, as shown in Figure 2b, spherical wave light sources with the same phase are assumed to be lined up on the same surface.

When a surface with protrusions or steps is the object to be measured, the spherical wave light sources are arranged with the phase given according to the shape of the object by setting the wavelength of the light source to $2\pi$ rad so that it corresponds to the height of the protrusions.

In this way, the measurement object is set up, with the projection shape expressed as a phase distribution.

In the simulation model shown in Figure 2a, the mesh size was set to 1/12 of the wavelength, which was confirmed to have no effect on the calculation results even if the mesh size was not chopped any finer, in order to set conditions where the mesh size does not affect the calculation results as much as possible, while considering the load on the computer memory. As a result, it was confirmed that even when the mesh size was set to 1/12 of the wavelength, the difference in calculation results did not change more than 1% from the results with a mesh size of 1/20. In addition, the arrangement of light sources as measurement objects was also set with 0.4 nm as the minimum unit interval, while considering the load on the computer's memory. To effectively use the limited memory available, the parallel side walls and the right-side wall of the computational domain were defined as perfectly matched layers (PMLs). The simulation model was designed to minimise the load on the memory capacity by defining the computational domain with the minimum possible memory size (2 TB), as in a previous study [25].

The light sources used in the simulations were plane-wave and spherical-wave light sources defined by Equations (1) and (2), which are derived from Maxwell's equation [2]

$$E_P = Va \times \exp\{i(2py/\lambda + \phi)\} \tag{1}$$

$$Es = Va \times \frac{\exp\left\{i(2p/\lambda\sqrt{x^2 + y^2} + \phi)\right\}}{\sqrt{x^2 + y^2}} \tag{2}$$

In this study, the spherical wave source was used as the model for the scattered light used in speckle interferometry. In the light source model, $Va$ is the electromagnetic field intensity, $\lambda$ is the wavelength, and $\phi$ is the initial phase of light from the source. As described before, the phase distribution was used to set the shape of the measurement object.

The simulation model assumed the objective used is a thin biconvex lens; the refractive indices of air and the lens were defined as 1.0 and 1.5, respectively. Furthermore, as shown in Figure 3a, the focal length (f = 37.8 μm) was specified by determining the lens focal point as the point where the highest electromagnetic field intensity is focused by the lens when plane waves as collimated light are irradiated from the left wall surface to the lens. Furthermore, when the spherical wave source (Pd) is positioned on the optical axis of the left wall surface, as shown in Figure 3b, and the lens is positioned at a distance from the left wall surface by the focal length of the lens defined in Figure 3a, the electromagnetic field intensity after passing through the lens is confirmed to be collimated light.

From these results, the focal length f of this optical system was confirmed to be 37.8 μm. In general, the lens used here is designed with a glass with a refractive index of 1.5 by means of arcs with a radius of 40 μm. Since both convex surfaces of the lens are formed by arcs of radius 40 μm, the focal length can be obtained as 40 μm if the thickness of the lens is sufficiently thin [1,2]. However, since the thickness of the lens is not necessarily thin enough, 5.83 μm in relation to the lens diameter, the focal length in this study was determined using the procedure shown in Figure 3. As a result, the NA of the objective lens

could be estimated as 0.37 [= $1 \times \sin(\tan^{-1}(15/37.8))$]. The diffraction limit as a Rayleigh criterion could then be obtained as 877 nm (=0.61 × $\lambda$/NA = 0.61 × 532/0.37).

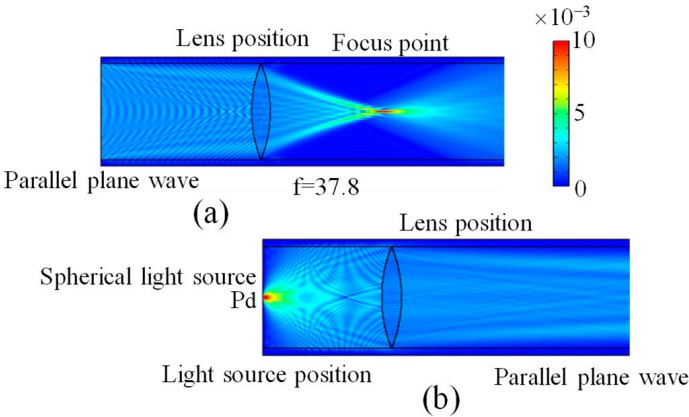

**Figure 3.** Focus length of the optical model. (**a**) Checking the focal point in the case of collimated light incidence. (**b**) Confirmation of collimated light at the focal point light source.

To further investigate the characteristics of the optical system, as shown in Figure 4, a spherical wave light source Pd (wavelength 532 nm) shown in Figure 3b was placed on the optical axis on the left wall of the optical system, and the focal length of the lens was set to 37.8 μm, as calculated in Figure 3.

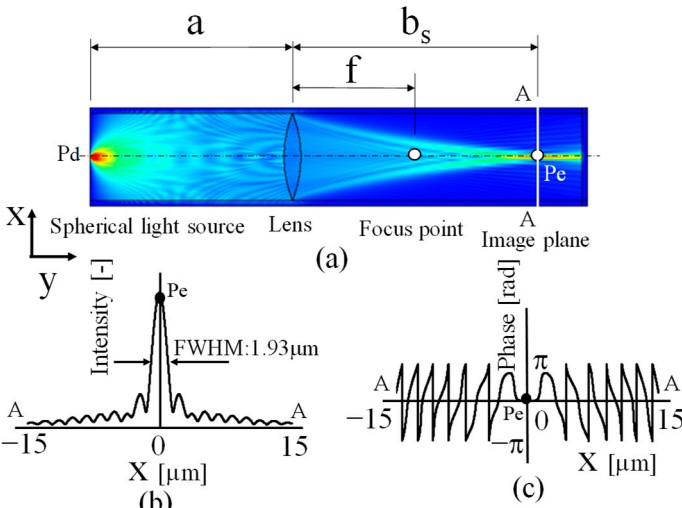

**Figure 4.** Optical system check when a light source is set on the light axis. (**a**) Electromagnetic field simulation result. (**b**) Intensity distribution in section A-A. (**c**) Phase distribution in section A-A.

The distance *a* between the lens and the light source and the distance $b_s$ between the lens and the point Pe where the electromagnetic field intensity is the highest near the image formation position after passing through the lens were examined based on the Gaussian lens formula [1,2].

In this study, the distance $b_s$ between the lens and the point Pe where the electromagnetic field intensity is the highest near the image formation position after passing through the lens was compared with the distance $b_c$ obtained based on the lens formula. The results calculated using the values of the distance *a* to the light source and the focal length *f* of the lens yielded the values as $b_c$ shown in Table 1.

**Table 1.** Factors of the optical system (µm).

| $a$ | $f$ | $b_c$ (Calculation) | $b_s$ (Simulation) | $b_c–b_s$ |
|---|---|---|---|---|
| 60 | 37.8 | 102.2 | 105.8 | −3.6 |
| 70 | 37.8 | 82.2 | 84.5 | −2.3 |
| 80 | 37.8 | 71.7 | 74.7 | −3 |

Comparing the value calculated based on the lens formula [1,2] with the distance between the point of highest intensity in the electromagnetic field intensity distribution obtained with computer simulation and the distance to the lens ($b_s$), it was confirmed that the values of $b_c$ and $b_s$ are approximately similar when $a$ is set to 60, 70, or 80 µm, considering the thickness of the lens. Based on these results, the computer simulation model set up in this study is considered to reproduce the actual optical system. Based on the results of the study, a focal length of 37.8 µm was used in the following simulation model. Furthermore, by setting the distance between the measurement object and the lens to 70 µm, the simulation was conducted using the $b_s$ (84.5 µm) values in Table 1 for the image formation position of the illumination light source. In this case, the lens magnification was 1.2×. Since this is a computer simulation, the phase distribution can be easily calculated using not only the intensity distribution of the light but also the real and imaginary parts of the intensity distribution.

In this case, the intensity distribution on the image plane (A-A in Figure 4a) is shown in Figure 4b and the phase distribution in Figure 4c.

On the imaging plane, an intensity distribution symmetrical in the x direction (Figure 4b) with a peak (Pe) on the optical axis can be confirmed. However, the phase distribution is obtained in the range of $−\pi$ to $\pi$ rad, since the calculation result as a simulation result is not phase-unwrapped as an inverse tangent function of the ratio of the real and imaginary parts of the intensity distribution. It can be confirmed that the phase is 0 rad at point Pe on the imaging plane, which is considered confocal for a spherical light wave source with an initial phase of 0 rad.

Using this computer simulation model that identifies the fundamental properties of the optical system, this study examines the physical effects of the Rayleigh criterion on the measurement results in a super-resolution technique based on speckle interferometry.

## 3. Results and Discussion

*3.1. Consideration of Rayleigh Criterion in Super-Resolution Technology Based on Speckle Interferometry by Simulation*

3.1.1. Consideration of the Case Where the Two Light Sources do Not Exceed the Rayleigh Criterion

The spherical wave light source located on the optical axis on the left wall of the model shown in Figure 4a was newly replaced on the left wall as a spherical wave light source symmetrical to the optical axis separated by 2 µm across the optical axis with the same phase. The electromagnetic field intensity distribution when light is emitted from the two light sources is shown in Figure 5a. The intensity and phase distributions on the imaging plane in this case are shown in Figure 5b,c.

The distance between the light sources was 2 µm, and the diffraction limit of the optics was 877 nm, which means that the two light sources are set at positions that do not exceed the Rayleigh criterion. As a result, spherical wave beams from two points 2 µm apart interfere, and Young's fringes [1,2] are formed in the intensity distribution, as is generally well known. In the B-B section of the imaging plane, as shown in Figure 5b, although the intensity distributions are not completely separated, the two peaks can be observed because they do not exceed the Rayleigh criterion. It can also be clearly observed that the zeroth-order and ±first-order light of the Young's fringes pass through the lens aperture and are focused at the image formation plane. The phase distribution on the imaging plane in this case is shown in Figure 5c.

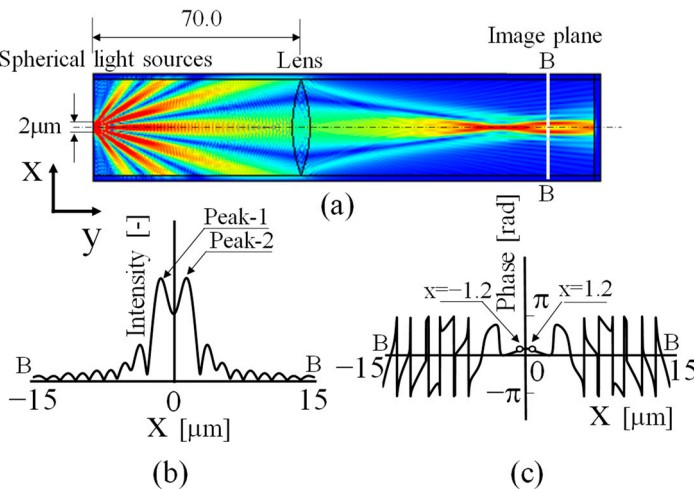

(a)

(b)

(c)

**Figure 5.** Simulated results when two light sources are set further than the Rayleigh criterion. (**a**) Electromagnetic field simulation result. (**b**) Intensity distribution in section B-B. (**c**) Phase distribution in section B-B.

When looking at the nearby optical axis in Figure 5c, the phase difference between the two light sources at x = −1.2 and x = 1.2 as the confocal where the light sources are set up was 0 rad, because the initial phases of the two spherical wave light sources set up in Figure 5a were both 0 rad.

That is, light from two spherical wave light sources with the same phase set at a distance that does not exceed the Rayleigh criterion can be considered not only as two light sources in terms of the intensity distribution on the image plane but also as the same phase in terms of phase distribution. This means that when considering the Rayleigh criterion in super-resolution technology based on speckle interferometry using coherent light as a light source, it is necessary not only to discuss the intensity distribution but also to investigate the phase distribution in detail, which has not been sufficiently investigated in the past.

3.1.2. Consideration of the Case where Light Sources Are Located at a Proximity Distance Exceeding the Rayleigh Criterion

Next, the case when the Rayleigh criterion in the earlier section is not exceeded occurred, and the distance between two spherical wave sources decreased from 2 μm specified in Figure 5 to 0.5 μm, as illustrated in Figure 6a.

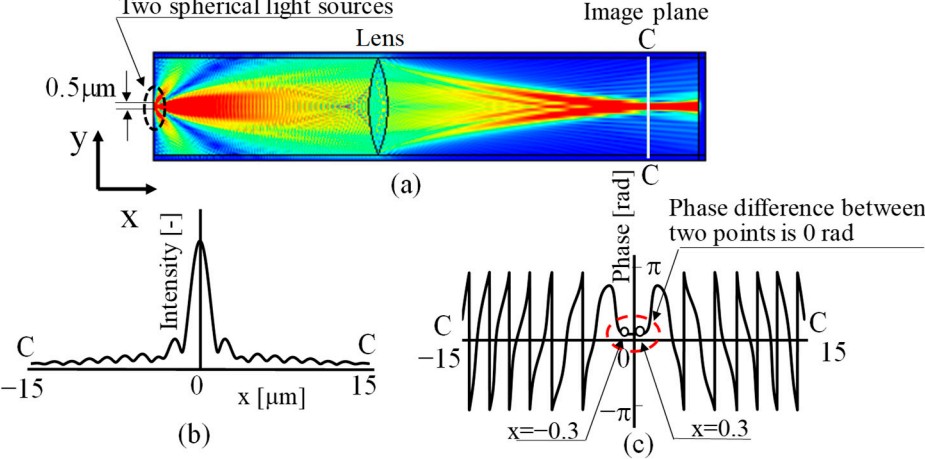

(a)

(b)

(c)

**Figure 6.** Simulated results when two light sources are set closer than the Rayleigh criterion. (**a**) Electromagnetic field simulation result. (**b**) Intensity distribution in section C-C. (**c**) Phase distribution in section C-C.

Figure 6a shows that as the distance between the two light sources becomes closer, the ±first-order light of the Young's fringes spreads out greatly to the left and right, clearly not passing through the lens aperture, and only the zeroth-order light passes through the lens and is focused at the image formation position. As a result, the intensity distribution on the imaging plane when emitting light from two points simultaneously, as shown in Figure 6b, cannot be observed as two separate light sources, even though the light was emitted separately from each of the two points. Instead, it is observed as a single peak with a maximum value near the centre between the peaks of the separately emitted lights. That is, the well-known phenomenon based on the Rayleigh criterion can be observed [1,2].

In this case, since the initial phase of the light source placed as a spherical wave source was set at the same value as 0 rad for both light sources, the phase difference of the observed light was 0 rad at the two points x = −0.3 and x = 0.3 close to the nearby axis surrounded by the red dashed line, as shown in Figure 6c.

It is considered that the phase difference corresponding to the initial phase of the two light sources is detected between the confocal points of the two light sources installed as the measurement object, as shown in Figure 5, on the image formation plane of the two light sources placed at a distance closer than the Rayleigh criterion. This means that when the diffraction limit based on the Rayleigh criterion is exceeded, the phase of the two light sources can be detected as the phase difference between the two points, although the intensity distribution can only be confirmed as a single point due to diffraction phenomena.

Thus, it can be understood that according to the traditional Rayleigh-criterion-based approach, imaging the shape of a measured object structure is a process based on intensity distribution, and therefore, due to phenomena caused by diffraction, it is not possible to observe microstructures beyond the diffraction limit using imaging techniques.

However, by treating the phase distribution and especially the phase difference between two light sources, it is possible to analyse the phase of light from each point, even if they are two points in close proximity, and there is a possibility that the shape of the measurement object can be reconstructed.

3.1.3. Consideration of Different Initial Phases of Light Sources Located at Close Proximity Distances Exceeding the Rayleigh Criterion

The difference between the experimental conditions based on super-resolution technology based on speckle interferometry and the simulation conditions when dealing with light from light sources of the same phase, as described in the previous section, is discussed next.

In the super-resolution technique based on speckle interferometry, when observing a microstructure, reflected light with a different phase is reflected from each point on the surface of the measured object, depending on the shape of the microstructure, and this reflected light is analysed.

However, the simulation in the previous section differs in that the light sources in close proximity have the same phase.

In this study, it was considered that the reflected light with different phases plays an important role in realising high resolution beyond the Rayleigh criterion in super-resolution technology based on the speckle interferometry technique.

Therefore, different from Figure 6, the initial phases of the two light sources were set as 0 rad and π rad, and the phase on the image formation plane was examined next when the distance between two points was set at 0.5 μm, as in Figure 6. The results are shown in Figure 7.

Comparing the electromagnetic field intensity distribution in Figure 7a with the result in Figure 6a, it can be seen that the phase of the intensity distribution reversed and the intensity of light near the optical axis weakened.

It can also be seen that the intensity of light in the diagonal directions, where the existence of intensity could not be observed in Figure 6a, became stronger. Since the initial phase differs by π rad between the two light sources, it is a natural result that the zeroth-order and ±first-order phases of Young's fringes in Figure 6a change by π rad.

As a result, light with strong intensity in the space between the zeroth- and ±first-order light in Figure 6a is considered to be generated, as shown in Figure 7a. In short, it can be understood that the separation of the two image points is due to destructive interference between the two images, as already suggested by microsphere-assisted micrococopy [16,17].

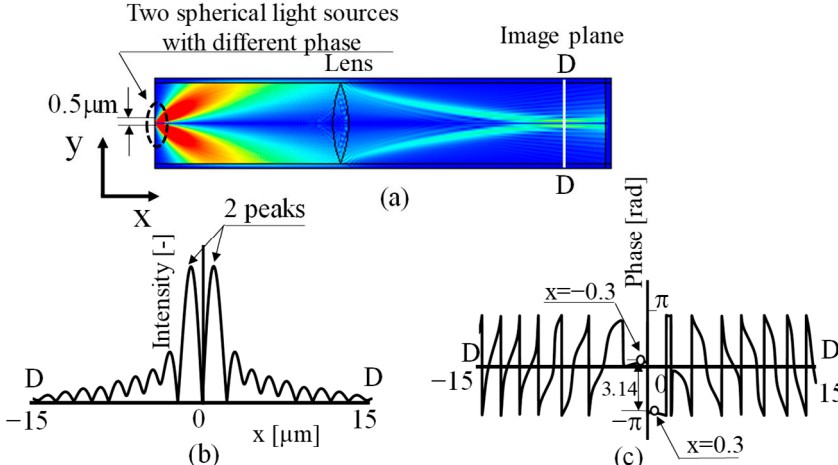

**Figure 7.** Simulated results when two light sources whose initial phases are different as π rad are set closer (0.5 μm) than the Rayleigh criterion. (**a**) Electromagnetic field simulation result. (**b**) Intensity distribution in section D-D. (**c**) Phase distribution in section D-D.

With this change in the direction of light, there is light that passes through the lens aperture, which did not exist in the in-phase case. This is thought to be the phenomenon of two bright spots on the image formation plane. In this case, when the light in this new direction is observed as an intensity distribution on the imaging plane, two intensity peaks can be observed near the optical axis, as shown in Figure 7b. Detailed observation of this phenomenon in Figure 7a shows that light emitted from the two points forms interference fringes known as Young's fringes and that the phase of the fringes is inverted and divided into two directions (upper and lower). Next, a part of the light from each of the two directions passes through the lens and reaches the image plane, forming two peaks as the intensity distribution. That is, Figure 7a,b confirms that there are two light points at the observation point on the image formation plane.

Furthermore, when observing the nearby optical axis of the phase distribution in Figure 7c in detail, the phase difference at the position of each white circle at x = −0.3 and x = 0.3, the confocal point of the two spherical wave light sources changed by π rad.

When the phase between the two light sources on the left wall differs by π rad, it can be confirmed that even if the two light sources are located beyond the Rayleigh criterion, the phase difference at the confocal point corresponding to the position of each light source set as the measurement object in the observed phase distribution differs by π rad. This means that even if the distance between two light sources exceeds the Rayleigh criterion, the phase difference between the light sources set up as light sources is preserved at the observation point.

This phenomenon suggests that the shape of an object can be measured beyond the diffraction limit as a phase distribution by detecting the phase at each point of the object with high resolution in super-resolution technology based on speckle interferometry technology. It can then be understood that for a phase distribution to exist, the existence of a geometrical unevenness distribution on the measured surface is required.

*3.2. Experimental Verification of a Phenomenon Obtained in Simulation Results That Occurs Based on a Phase Change between Two Light Sources Located beyond the Diffraction Limit*

In the simulation, it was shown that when the phases of two light sources in close proximity beyond the diffraction limit are different, the existence of the two light sources

can be confirmed by detecting the phase difference between the two light sources if the light is coherent. In other words, if two light sources are based on the diffraction limit, which was thought to be impossible to confirm the existence of two points that exist beyond the Rayleigh criterion, based on the simulation results, the separation of the existence of two light sources beyond the diffraction limit is considered possible by detecting the phase difference between the two points with high resolution. Therefore, it was experimentally verified whether the phenomena based on the simulation results could occur in reality using a real optical model that was simplified as much as possible.

In the optical system used in the experiment, the diffracted image shown in Figure 8a with a circular aperture, formed by a laser light source with a source wavelength of 532 nm, was used as the diffracted image model [2] when the light source was observed using a circular lens. Two diffraction image models were prepared with light emitted from the same laser and with no phase difference between the two diffraction image models, and the two models were superimposed so that they overlapped from the left and right.

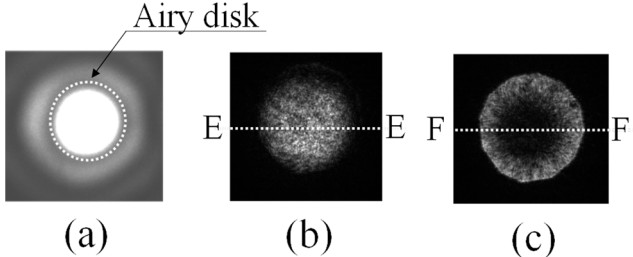

**Figure 8.** Overlap of diffracted lights with phase information. (**a**) Diffracted light source model. (**b**) Overlap of two light sources with phase difference 0 rad. (**c**) Overlap of two light sources with phase difference $\pi$ rad.

In this case, if the overlap exceeds the Airy disk, the light source is considered only one light point. The general situation regarding diffraction limits based on the well-known Rayleigh criterion [2] arises. Furthermore, when the two lights are superimposed so that they gradually coincide spatially, the interior of the Airy disk is observed as a single bright light source because the phases of the two lights are originally equal, as shown in Figure 8b. This is also a well-known phenomenon that generally occurs when dealing with the Rayleigh criterion [2].

In this case, the E-E section of the intensity distribution in Figure 8b is shown in Figure 9a. It can be observed that the entire inner surface of the Airy disk is brightened.

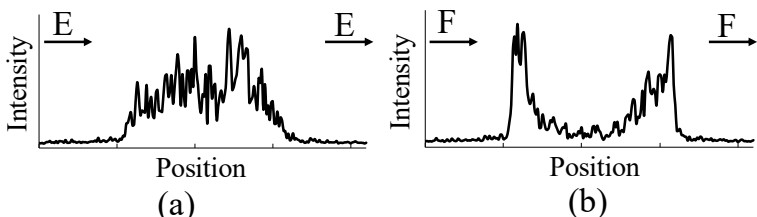

**Figure 9.** Section of intensity. (**a**) Two light sources with phase difference 0 rad. (**b**) Two light sources with phase difference $\pi$ rad.

Next, the phase difference between the two overlapping lights was changed to $\pi$ rad by changing the optical path length of one of the lights. It could be confirmed that the bright state inside the Airy disk, as shown in Figure 8b, changes to a dark state at the centre of the light, as shown in Figure 8c. In this case, the intensity distribution in the F-F cross section of Figure 8c is shown in Figure 9b. Clearly, the central area becomes darker and the peripheral area becomes brighter, just as in Figure 7b, which was observed in the simulation.

Thus, if the two light sources in close proximity beyond the Rayleigh criterion are coherent and the phases of the two light sources are different, experimental results confirm

that the two light sources can be separated by detecting the phase difference between the two points on the image formation plane, as in the simulation results.

### 3.3. Influence on the Detection Phase of two Coherent Light Sources with Different Phases as the Distance between them Changes

As shown in this study, it was found that even two light sources exceeding the Rayleigh criterion can be observed as two points using phase analysis of coherent light.

In Figure 7, two light sources 0.5 µm apart were observed. Therefore, the next case in which the two light sources are even closer to each other was discussed.

The results for the case where the distance between the two light sources is 0.25 µm are shown in Figure 10. Figure 10a–c shows the results when the two light sources have the same phase (0 rad). Similar to the results shown in Figure 6, the two light sources placed beyond the diffraction limit cannot be separated. In the intensity distribution in Figure 10b, two light sources cannot be considered as two light sources. However, in the phase distribution in Figure 10c, it can be clearly confirmed that the phase difference between the two light sources is 0, as shown by the red dashed line.

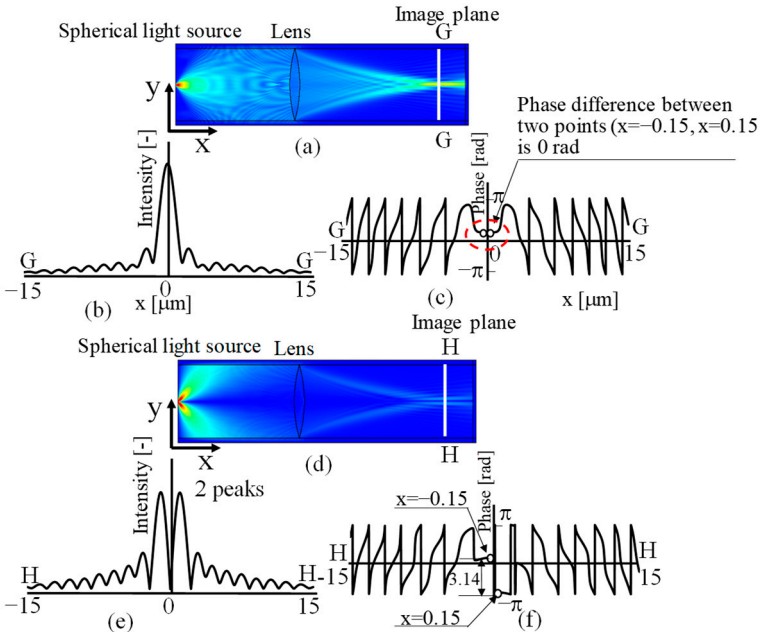

**Figure 10.** Simulated results with two light sources in close proximity to 0.25 µm, exceeding the Rayleigh criterion when the initial phases of the two light sources are in the same phase and differ by $\pi$ rad. (**a**) Electromagnetic simulation results for the same phase. (**b**) Intensity distribution for the same phase in section G-G. (**c**) Phase distribution for the same phase in section G-G. (**d**) Electromagnetic field simulation results differ by $\pi$ rad. (**e**) Intensity distributions differ by $\pi$ rad in section H-H. (**f**) Phase distributions differ by $\pi$ rad in section H-H.

In contrast, in Figure 10d–f, where the phase difference between the two light sources changes to $\pi$ rad, two peaks can be observed in the intensity distribution in Figure 10e.

Furthermore, in the phase distribution in Figure 10f, it can be confirmed that the phase difference between the two points changes by $\pi$ rad at the confocal point where the light source is located.

These results show that in an optical system with a diffraction limit of 877 nm, if two light sources 250 nm apart are coherent light sources and their phases are detected, it is possible to observe them as two light sources beyond the diffraction limit.

Furthermore, how an observation becomes possible when two light sources are in close proximity was investigated using simulation.

First, the light source wavelength was 532 nm, and the phase difference between the two light sources was set as $\pi$ rad when the distance between the two light sources varied from 0.01 to 0.5 times the light source wavelength ($\lambda$).

In Figure 11, the horizontal axis is the distance between the two light sources (w) and is given as a multiple of the wavelength $\lambda$. The vertical axis is the detected phase difference between the two light sources.

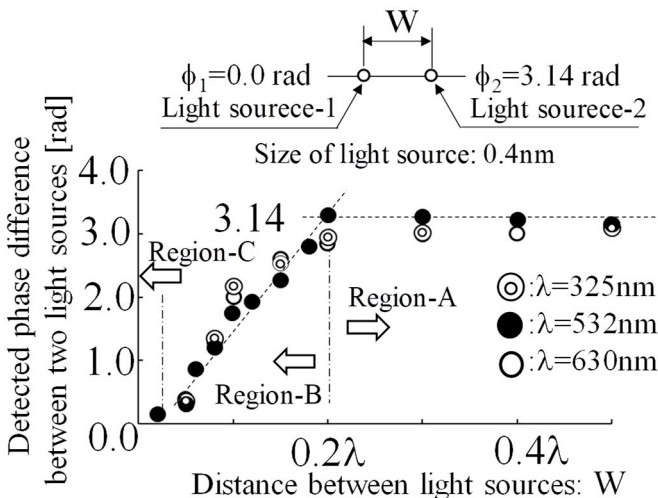

**Figure 11.** Relationship between the distance of light sources and the detected phase.

In the result for the wavelength of 532 nm indicated by the black circle ($\bullet$), it can be observed that the phase of two light sources in close proximity can be detected as $\pi$ rad, which is set as the phase difference, up to about 0.2 $\lambda$ (actual length: about 100 nm). Next, as the distance becomes closer than 100 nm (0.2 $\lambda$), the phase difference gradually becomes increasingly smaller, and even if the phase difference is set as $\pi$ rad, it can no longer be detected as $\pi$ rad.

However, even if it is no longer possible to accurately detect dimensions related to the shape of the object, it is still possible to observe the approximate shape of the measured object. For example, in the experimental results of a previous report [26], although the groove depth of a 100-nm-wide groove could be detected almost accurately when observing a 100-nm-wide groove, the groove depth of a 60-nm-wide groove could not be accurately detected as the actual groove depth, although it could be captured as a groove.

Thus, in the observation of microstructures using speckle interferometry, there are several measurement limits in the experimental measurement process, such as the range where dimensions can be accurately measured (e.g., Region-A in Figure 11), the range where dimensions cannot be accurately measured but shapes can be captured (e.g., Region-B in Figure 11), and the range where the state of the measurement is not yet clearly understood (e.g., Region-C in Figure 11).

It is thought that there are several levels of measurement limits.

Therefore, it is necessary to investigate in detail the regions below 0.2 $\lambda$ in Figure 11 (Region-B and Region-C in Figure 11) in the future. Furthermore, based on the results of this study, it is also necessary to discuss the causes of why such regions occur. Based on the results of these further investigations, the measurement limits of this method should be considered in more detail.

In this study, simulations were performed on the basis of the experimental results already reported. As a result, the wavelength was considered 532 nm. However, as a general concept in optical measurement, it is important to know how a change in wavelength affects the measurement results. Therefore, next, a simulation was performed to see how a light source with a different wavelength, as well as 532 nm, affected the measurement results.

For the light source wavelength, it was decided to consider visible light lasers, which are commonly used for measurements. The case of a long wavelength of 630 nm, modelled after a He-Ne laser (wavelength: 632.8 nm), is indicated by a white circle (○) in Figure 11.

The results are also shown in Figure 11 as double white circles (◎) for the case of the 325 nm wavelength modelled after the shorter wavelengths of He-Cd lasers (wavelengths: 325 nm and 442 nm).

For different light source wavelengths in Figure 11, it can be found that the phase difference between the two light sources can be accurately detected up to approximately one-fifth of the wavelength (0.2 λ), regardless of the wavelength. This indicates that even when the diffraction limit is exceeded, observation of finer structures becomes possible as the light source wavelength becomes shorter.

These results show that in the observation technique for structures with geometries beyond the diffraction limit using speckle interferometry, the phase difference between two nearby light sources is preserved during detection, even between two points beyond the diffraction limit, when coherent light is used.

By using this phenomenon, it is thought that super-resolution beyond the Rayleigh limit, which has been thought to be undetectable using the conventional Rayleigh criterion based on incoherent light, is realised by detecting the phase difference at each position in the microstructure observation technique based on speckle interferometry technology.

The discussion in this study also focused on super-resolution technology during image sampling, particularly for optical observation of microstructures. However, the sensing technology obtained in this study, which reveals the possibility of realising super-resolution based on phase manipulation of light waves with coherent properties, could also be applied to other sensing fields using electromagnetic waves, such as radar sensor technology [28].

In the future, the results of this research may lead to the use of phase manipulation technology in sensing related to super-resolution using electromagnetic waves with coherent properties, not only in the field of optics, but also in a wide range of other fields.

## 4. Conclusions

In this study, a physical explanation for the super-resolution phenomenon in a new microstructure observation technique using speckle interferometry [22], in which the realisation of super-resolution has been experimentally confirmed, was discussed. In this explanation, a computer simulation was used to investigate why the observation of microstructures exceeding the Rayleigh criterion, which had long been considered unexceedable, could be realised.

The simulation results show that when coherent light is used as a light source and the phase difference between two light sources is different, the phase difference is preserved at the image formation position at the time of detection, even if the two light sources are close to each other beyond the Rayleigh criterion.

By using this physical phenomenon, it was shown that light from two points in proximity exceeding the Rayleigh criterion can be detected as light from two points in proximity exceeding the Rayleigh criterion by capturing the phase distribution, although it was previously thought that light from two points in proximity exceeding the Rayleigh criterion cannot be separated on the basis of incoherent light.

Furthermore, regarding the measurement limit of super-resolution technology based on the speckle interferometry technique, the simulation model used in this study clarified that the phase difference between two light sources can be accurately detected up to a distance as close as about 20% of the light source wavelength. In the discussion that led to this conclusion, it became clear that this super-resolution technology has three types of measurement limits: (1) the range where dimensions can be accurately measured, (2) the range where dimensions cannot be accurately measured but shapes can be captured, and (3) the range where it is difficult to accurately measure differences in the object's steps, etc. In addition, it was confirmed that the discussion of these measurement limits is consistent with the results of the previous experiments.

**Author Contributions:** Conceptualization, Y.A.; methodology, Y.A.; software, Y.A. and T.C.; validation, Y.A.; formal analysis, Y.A and T.C.; investigation, Y.A. and T.C.; resources, Y.A.; data curation, Y.A. and T.C.; writing—original draft preparation, Y.A.; writing—review and editing, Y.A.; supervision, Y.A.; project administration, Y.A.; funding acquisition, Y.A. All authors have read and agreed to the published version of the manuscript.

**Funding:** This research was funded by JSPS KAKENHI (grant number 20H02165).

**Institutional Review Board Statement:** Not applicable.

**Informed Consent Statement:** Not applicable.

**Data Availability Statement:** Not applicable.

**Acknowledgments:** We would like to thank Dahai Mi of Keisoku Engineering System CO., LTD, for his kind guidance and support in the use of COMSOL Multiphysics.

**Conflicts of Interest:** The authors declare no conflict of interest.

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
