# Peer review of "Simulation-Based Considerations on the Rayleigh Criterion in Super-Resolution Techniques Based on Speckle Interferometry"

_photonics, doi:10.3390/photonics10040374_

Round 1

Reviewer 1 Report

Fig.1(c) in image plane PC’ should be replaced by P’C to conform with the text.

Fig. 3(b) Sphere wave should be changed to Spherical wave.

Page 11, paragraph 2, line 2---coherent light should be replaced by coherent (delete light).

Please find details in the attachment file.

Reviewer 2 Report

Paper: Simulation-based considerations on the Rayleigh criterion in super-resolution techniques based on speckle interferometry from Yasuhiko Arai and Tong Chen in MDPI Photonics

In this paper authors consider theoretically and experimentally the image of two close point-sources and how the phase distribution along an object can contribute to improve the resolution beyond the Rayleigh criterium when using coherent light. This is used to explain super-resolution in speckle imaging techniques. The paper is very well explained but does not cite papers where this phenomenon was already described. The more original result is figure 12.

1-The main result, imaging two close point-sources with a microlens taking into account their phase difference to explain a resolution improvement, has been already described in these papers:

A.V. Maslov, V.N. Astratov. Appl. Phys. Lett., 108, 051104, 2016
I. Kassamakov, S. Lecler, A. Nolvi, A. Leong-Hoi, P. Montgomery, E. Haeggstrom, Sci. Reports 7, 3683, 2017

Moreover, references [1][2][3] to describe the Abbe limit is perhaps too much for a well-known basic limitation. Also 7 papers out of 20 are from the authors, it is generally considered as too much auto-citation.

2-When the authors talk about the other super-resolution optical techniques: near-field optics (SNOM or microsphere-assisted microcopy) and superlens (as the work of Pendry) may be cited.

3-In figure 5, it will be useful to give value of the Full-Width at Half Maximum of the peak.

4-In section 3.1.3, page 9, line 51, authors say “In short, it can be understood that the direction of light emission changes with the change in phase between the two light sources”. It is correct, must simply speaking the separation of the two image points is due to destructive interferences between the two images. It must be written.    

5-Other comments:
-Page 2, the section lines 47-52 is not clear and must rewritten.
-Figure is a too basic simulation. It doesn’t need being in the paper. Formulae must be rewritten.
- in the formula 0.37 [= 1 × sin (tan-1 (15/37.8))]        -1 is not correctly written.
- when the parameters “a” and “bs” are written in the text, they must be in italic. “s” is in index.  

Reviewer 3 Report

This paper presents simulated verification of the super-resolution method in optical imaging.The innovation is attractive. The paper is well-organized, and the experiments are sufficient However, I think that authors should add more results and comments before it can be recommended for publication:

(1) The super-resolution result is indeed obtained for two single target. However, from the practical view, what we concern more is the surface target. For example, an optical photo containing human beings, animals and so on. In this case, is the proposed phase-modulated method feasible?

(2) In a variety of applications, the coherent light is difficult to obtain. For example, the passive optical sensing. In this case, how to real super-resolution?

(3) There are also a lot of signal processing method to realize super-resolution. For example, the classical Lucy-Richardson method,

[1] W.H Richardson. “Byesian-based Iterative Method of Image Restoration.,”Journal of the Optical Society of America, 62(1),pp:55-59, 1972

[2] L.B Lucy. “An Iterative Technique for the Rectification of Observed Distributions,” The Astronomical Journal, 79(6),pp:745-754, 1974

Authors should increase the relevance.

(4) The super-resolution problem also occurs in other kinds of sensors, such as the radar sensor,

[1] Zhang Y, Luo J, Li J, et al. Fast inverse-scattering reconstruction for airborne high-squint radar imagery based on Doppler centroid compensation[J]. IEEE Transactions on Geoscience and Remote Sensing, 2021, 60: 1-17.

is the conclusion in this paper also applicable? Authors should increase the relevance and make comments.

(5) Super-resolution does not only means separating the two sources, but also means estimating the true location of them. Therefore, the estimation error on location should be also evaluated.

Round 2

Reviewer 2 Report

The paper has been improved, however there is still one big issue: 

the references [16] Maslov 2016 and [17] Kassamakov 2017 have been added but as an example of super-resolution. It is not the main raison why they must be cited. In these two papers, it is demonstrated that the phase response of the sample can contribute to increase the resolution beyond the diffraction limit, what is the aim subject of the present paper (The authors have to look Figure 4 in ref [16] and Figure 2 in ref [17]).

Therefore, page 10, when the authors say “In short, it can be understood that the separation of the two image points is due to destructive interference between the two images”, they can add “as it was already suggest for microsphere assisted microcopy [16,17]”

Figure 3 is not mandatory.
